# Specificities of the DMD Gene Mutation Spectrum in Russian Patients

**DOI:** 10.3390/ijms232112710

**Published:** 2022-10-22

**Authors:** Elena Zinina, Maria Bulakh, Alena Chukhrova, Oksana Ryzhkova, Peter Sparber, Olga Shchagina, Aleksander Polyakov, Sergey Kutsev

**Affiliations:** Research Centre for Medical Genetics, Moskvorechye St., 1, 115522 Moscow, Russia

**Keywords:** DMD, Duchenne/Becker muscular dystrophy, selective screening program, mutation spectrum, DNA-diagnostics

## Abstract

Duchenne/Becker muscular dystrophy (DMD/BMD) is the most common form of muscular dystrophy, accounting for over 50% of all cases. In this regard, in Russia we carry out a program of selective screening for DMD/BMD, which mainly involves male patients. The main inclusion criteria are an increase in the level of creatine phosphokinase (>2000 U/L) or an established clinical diagnosis. At the first stage of screening, patients are scanned for extended deletions and duplications in the DMD gene using multiplex ligase-dependent probe amplification (MLPA SALSA P034 and P035 DMD probemix, MRC-Holland). The second stage is the search for small mutations using a custom NGS panel, which includes 31 genes responsible for various forms of limb-girdle muscular dystrophy. In a screening of 1025 families with a referral Duchenne/Becker diagnosis, pathogenic and likely pathogenic variants in the DMD gene were found in 788 families (in 76.9% of cases). In the current study, we analyzed the mutation spectrum of the DMD gene in Russian patients and noted certain differences between the examined cohort and the multi-ethnic cohort. The analysis of the DMD gene mutation spectrum is essential for patients with DMD/BMD because the exact mutation type determines the application of a specific therapeutic method.

## 1. Introduction

Duchenne/Becker muscular dystrophy (DMD/BMD) is an X-linked hereditary neuromuscular disorder caused by mutations in the dystrophin gene [1]. There are two distinct allelic forms of the disease with different symptom severity. The most common and severe type—Duchenne myodystrophy (DMD) (OMIM: 310200)—manifests at the age of one to three years and is characterized by progressive proximal muscular weakness [2]. Most patients lose the ability to move independently before puberty. Aside from that, 98% of patients have exacerbations in the form of cardiomyopathy with subsequent development of respiratory and cardiac deficiency [3,4]. Becker muscular dystrophy (BMD) (OMIM: 300376) is a milder allelic form of the disorder. Clinical manifestations are observed at the age interval of 15 to 20 years with varying severity, from muscular weakness to total inability to move independently at the end of the second decade of life [5,6]. It is known that DMD and BMD in aggregate affect one per 3500–5000 newborn males, and in rare cases the disease can affect females as well [7]. According to this data on DMD/BMD prevalence, the estimated number of patients with DMD/BMD in the Russian Federation is approximately 4000 [1].

DMD/BMD is caused by mutations in the DMD gene, which is located in the Xp21.2-p21.1 region. The DMD gene encodes the dystrophin protein, which is a structural cytoplasmic protein and a part of the dystrophin-associated glycoprotein complex (DGC). The main function of DGC is to stabilize the muscle cell membrane and protect it from injury induced by muscle contraction. In the absence of a functional DGC, cell membranes are destabilized, destroyed, and, as a result, muscle fiber atrophy occurs [8].

The dystrophin gene is one of the largest protein-coding genes in the human genome, consisting of 79 exons and reaching approximately 2.6 Mb in length [9]. Mutations in this gene lead to a deficit of various isoforms of the dystrophin protein, which is expressed in many tissues and organs (skeletal, cardiac, and smooth muscles, central nervous system, retina, lungs, kidney, liver). However, most of mutations in DMD gene affect the expression of the muscle isoform (Dp427-M) [8]. The major determinant of differences in clinical symptoms of DMD and BMD is the reading frame rule. DMD is caused by mutations giving rise to an mRNA with disrupter reading frame, which can lead to nonsense-mediated RNA decay. As the result of this is the total loss of dystrophin synthesis. BMD is usually caused by mutations not leading to a frameshift, resulting in the formation of a truncated but functionally active protein [1,10,11]. This frameshift rule explains the phenotypical differences between patients with DMD and BMD. Approximately 90% of mutations are estimated to correspond to this rule [6].

According to the data presented by Annexstad E.J. et al., more than 4700 different mutations are identified in the DMD gene [7]. The most common mutations in the case of DMD/BMD are gross deletions and duplications of one or several exons [12]. They comprise 55–65% and 6–11% of all mutations in this gene, respectively [13]. These mutations may appear in any part of the gene, but it is worth noting that there are “hot” regions in the DMD gene that are more susceptible to deletions and duplications: on the 5′-terminus (exons 2–19) and in the distal half of the central rod domain (exons 44–53) [14]. Other DMD/BMD cases are caused by point mutations (approximately 20–30%), including missense (0.4%), nonsense (10.2%), and splice site mutations (2.8%), as well as minor rearrangements (insertions/deletions) (6.9%) [1]. Less than 1% of patients may also have deep intronic mutations, which significantly affect splicing [15]. The mutation spectrum of the dystrophin gene varies greatly depending on the population, which could be explained by the role of ethnic origin in mutagenesis [16]. In the current study, we analyzed the mutation spectrum of the DMD gene in Russian patients and noted certain differences between the examined cohort and the multi-ethnic cohort.

The analysis of the DMD gene mutation spectrum is essential for patients with DMD/BMD because the exact mutation type determines the application of a specific therapeutic method. As of now, two possible treatment approaches exist. The first one is therapy based on skipping certain exons of the gene, such as 45, 51, 53 [17], which allows us to turn a frameshift deletion into a non-frameshift deletion, leading to a milder clinical course of the disease in the patient. The second approach is based on premature stop codon skipping. These methods allow us to decelerate the disease progression, through preserving independent movement for as long as possible and extending the period until the upper limb function loss.

## 2. Results

As a result of selective screening, pathogenic and likely pathogenic variants in the DMD gene were detected in 788 (76.9%) out of 1025 comprehensively examined families. In 9.2% of cases (94 families), we detected variants in genes and mutations, which led to various types of limb-girdle muscular dystrophy. In 143 (13.9%) cases, we could not identify causative variants.

We analyzed the mutation spectrum of the DMD gene in the cohort of DMD/BMD patients with a confirmed molecular genetic diagnosis (788 probands). The distribution of mutation types in Russian DMD/BMD patients is shown in Figure 1. Gross deletions and duplications were detected in 49.0% (386/788) and 14.5% (114/788) of patients, respectively (Figure 1A). During the result analysis, we also evaluated the effect of the mutation on the reading frame, and in the majority of cases the detected variants led to a frameshift, causing a more severe phenotype.

Minor mutations comprised approximately 36.5% (288/788) of all mutations in our cohort. It is worth noting that among minor mutations, the most frequent ones were nonsense mutations—19.3% (152/788), which makes Ataluren treatment viable for a significant number of patients. Aside from that, we detected minor deletions/duplications/insertions, leading to a frameshift (81/788, 10.3%), splice site mutations (43/788, 5.4%), missense variants (8/788, 1.1%), and deep intronic variants and minor in-frame deletions—0.2% (2/788) each (Figure 1B).

### 2.1. Deletions

Using multiplex ligase-dependent probe amplification (MLPA) analysis, we established that out of 788 non-related patients with mutations in the DMD gene, 386 (49.0%) had gross deletions spanning one or several exons of the gene. In 5 patients out of these 386, partial deletions of one exon were detected using alternative methods, such as mass parallel sequencing and Sanger sequencing. Single-exon deletions were detected in 66 (17.1%) cases out of 386. The most common deletions involved exons 1, 44, 45, 50, 52 (42/386, 10.8%).

Deletions in 386 non-related probands were represented by 158 various exon deletions (Figure 2), which spanned from one exon to the entire DMD gene. The data on the deletion distribution confirm that they most frequently appear in the medial-distal region of the central domain, exons 44 to 55, as well as on the 5′-terminus of the N-domain (exons 2 to 20) [14]. The highest deletion concentration was in exons 44 to 55, which was detected in approximately 240 (62.2%) patients with deletions (Figure 2). The most common was the three-exon deletion (48–50), which was detected in 16 patients. Aside from that, 105 (27.2%) patients had deletions in exons 2 to 20, and the most common of them involved exons 3 to 7 and was detected in 8 patients.

It is also worth noting that the regions least susceptible to deletions were exons 21–44 and 57–79; more than half of these deletions were detected only once. Deletions in these regions comprised only 10.6% (41/386) of all detected deletion cases.

### 2.2. Duplications

As a result of selective screening, we established that duplications comprise 14.5% (114/788) of all mutations in the DMD gene, representing 78 different duplication types (Figure 3). In contrast to deletions, duplications were distributed more evenly throughout the entire gene; many of them were detected only once. In accordance with previously published studies, duplication of exon 2 was the most common [18]. In our cohort, it was detected in 11 patients. Aside from that, according to White S.J. et al., another common duplication spanned exons 3–7 [19]. In our cohort, it was detected in six patients. The rest of the duplications were distributed almost evenly throughout the entire gene, but the majority of them were localized in the 5′ region (approximately 54% (64/114) of duplications). The distribution of gross deletions and duplications in the DMD gene in the cohort of Russian patients is presented in Figure 4.

In addition, four DMD patients had complex rearrangements: del14–20, 27; del56–57, 61–62; dup52–60, 63–79; and dup5–6, del7. These variants were not described in the HGMD and LOVD databases. At the moment, the mechanism of these rearrangements has not been studied in detail.

Figure 4 implies that the distribution of gross deletions and duplications is not accidental: there are two “hot” regions, which corresponds with the data from previous studies on other cohorts [15,20].

### 2.3. Minor Mutations

The spectrum of detected minor mutations is presented in Table 1. In total, minor mutations were detected in 288 cases out of 788 (36.5%), out of which nonsense mutations comprised 19.3% (152/788), frameshift mutations—10.3% (81/788), splice site mutations—5.4% (43/788), missense mutations—1.1% (8/788). We detected minor in-frame deletions, as well as intronic variants, twice each in the examined cohort.

Among the 228 detected minor mutations, 109 were not previously described as pathogenic or registered in the LOVD, HGMD, or ClinVar databases. The clinical significance of all registered and novel variants was evaluated, according to the Russian variant interpretation criteria from the “Guide to interpretation of human DNA sequence obtained with mass parallel sequencing (MPS)” [21]. The main criteria of pathogenicity for the nucleotide sequence variants were: PVS1—LOF variants leading to the absence of protein (nonsense mutations, frameshift mutations, canonic splice site nucleotide alterations); PS1—nucleotide sequence variants changing to the same amino acid in the same position as a variant previously described as pathogenic; PM2—variants not present in the GnomAD control cohort or present with an extremely low frequency; PM5—novel missense variants, leading to an amino acid change in the same position as previously described pathogenic missense variants; PP3—variants evaluated as pathogenic by at least three in silico prediction programs.

The clinical significance of the detected nucleotide sequence changes was bioinformatically analyzed using on-line predictors—Mutation Taster (http://www.mutationtaster.org/ (accessed on 18 September 2022)), SIFT/Provean (http://provean.jcvi.org/index.php (accessed on 18 September 2022)), PolyPhen-2 (http://genetics.bwh.harvard.edu/pph2/index.shtml (accessed on 18 September 2022)), Human Splicing Finder (http://www.umd.be/HSF/ (accessed on 18 September 2022)), NetGene2 (http://www.cbs.dtu.dk/services/NetGene2/ (accessed on 18 September 2022)), ASSP (http://wangcomputing.com/assp/ (accessed on 18 September 2022)), FSplice (http://www.softberry.com/berry.phtml?topic=fsplice&group=programs&subgroup=gfind (accessed on 18 September 2022)).

According to multiple studies, the “hot regions” of minor mutations were not identified; all variants were distributed almost evenly throughout the entire DMD gene [18,22], which also corresponds to our data. However, in Russian patients, minor mutations most frequently occurred in exon 70 (11.1%, 32/288), although this exon is slightly shorter (137 bp) than the average exon in the DMD gene (approximately 175 bp) [23].

### 2.4. Other Genes

During the examination of the DMD/BMD cohort using a panel consisting of 31 genes, causative nucleotide sequence variants in the following 16 genes were identified in 95 (9%) patients from 94 families: CAPN3—3.3%, ANO5—1.2%, FKRP—1.0%, POMT1—1.0%, LMNA—0.9%, SGCA—0.7%, EMD—0.3%, DYSF, SGCB, SGCD, SGCG, GNE, GAA, CRPPA, POMGNT1, CAV3 < 0.1% each. This panel consists of genes and mutations, which lead to various types of limb-girdle muscular dystrophies and allows us to carry out differential diagnostics and to increase the molecular genetic examination informativity.

## 3. Discussion

Currently, the main aim of selective screening is early and distinctive molecular genetic diagnostics of DMD/BMD result in the detection of patients suitable for the available therapy types.

Patients are being referred for genotyping from 77 regions of the Russian Federation. The diagnostic algorithm includes the exon copy number MLPA analysis with subsequent minor mutation detection using massive parallel sequencing.

### 3.1. Mutation Types

As a result of selective DMD/BMD screening, we identified pathogenic and likely pathogenic DMD variants in 788 non-related patients. The majority of variants were represented by gross deletions of one or several exons (49.0%), as well as single-nucleotide nonsense mutations (19.3%). Gross duplications (14.5%) and various minor mutations were also detected frequently in the cohort. Approximately half of the variants identified in the current study were not previously described.

When comparing the DMD mutation spectrum with the published data, we noted that Russian DMD/BMD patients frequently have minor mutations (36.5%), especially nonsense—(19.3%) and minor frameshift mutations (10.3%), while gross deletions are less frequent (49%) than in studies from other countries (65–70%) (Table 2). The percentage of duplications in the current study is slightly higher (14.5%) than in the cohorts registered in various databases (8.0–11.0%) (Table 2). Meanwhile, the prevalence of missense variants and intronic mutations in the compared cohorts differs insignificantly. The fact that the differences in intronic variant prevalence are statistically insignificant may be caused by the small number of detected mutations of that type. This is due to the fact that current DMD diagnostic tools the intronic rearrangements are not possible to detect. (Table 2).

It is also worth noting that the proportion of mutations in the examined cohort matches the data presented by Flanigan K.M. et al. (USA) [18] and Takeshima Y. et al. (Japan) [27]. These studies were published in 2009 and 2010, respectively. The cohort sizes were 1111 families in the USA study and 442 families in the Japanese study. The rate of deletions (42.9%) in USA differs greatly from the worldwide results. Flanigan K.M. et al. have made an assumption that this percentage is lower because of a bias in patient selection: for a long time, DMD minor mutation detection was not widely available in USA; therefore, at the time of that study, the described cohort was overloaded with minor mutations. The differences can also be observed in the cohort presented by Japanese researchers: similarly to the current study, the proportion of minor mutations exceeds the estimated 20.0% and reaches 30.0%. The first assumption made by Japanese scientists is that the difference might have been caused by their approach to Duchenne/Becker myodystrophy diagnostics: the use of dystrophin mRNA analysis for minor mutation detection allows us to identify a large number of previously overlooked mutations, including intronic. The second explanation of these differences between cohorts presented by various countries is the ethnic diversity of mutagenic factors, especially the speed of methylated cytosine deamination (according to Juan-Mateu J. et al., a significant number of nonsense mutations are C > T replacements) [26] and the specificities of chromatin structure [27]. Additionally, as previously stated, the heterogeneity of mutation spectrum in other populations supports the existence of ethnic differences and ethnicity may play a role in mutation diversity [16]. Selvatici R. et al. have done extensive research looking at ethnicity-related DMD genotype landscapes in 12 countries. As a result of this work, a variety of mutation types was identified in different countries, which makes it possible to assess the suitability of personalized therapy in a particular country. Ultimately, this may affect diagnostic approaches [28].

At the moment, it is quite difficult to determine the exact cause of mutation spectrum differences between the current study and the worldwide results. It is worth noting that for a long time, the DMD/BMD diagnostics in the Russian Federation was limited to gross rearrangement detection using multiplex amplification of exons most frequently affected by deletions in patients. Around 2015, the Research Centre for Medical Genetics implemented minor mutation detection using massive parallel sequencing. Both factors mentioned above might be the cause of the relatively high rate of minor mutations in the examined cohort. However, it is impossible to completely exclude the effect of population differences on the mutation spectrum.

### 3.2. Gross Rearrangements

According to the published data, the DMD mutation spectrum is mostly represented by gross rearrangements spanning one or several exons. According to the data presented initially, in approximately 55–65% of cases, researchers detect gross deletions [29], and in 6–11% of cases—gross duplications [14]. We compared the distribution rates of various mutation types in France [24], China [25], USA [18], Spain [26], Italy [16], and Japan [27], as well as in the multi-ethnic cohort presented in the TREAT-NMD DMD Global register [1]. In the majority of cohorts, the rates of deletions and duplications correspond to the published results—5–65% and 6–11%, respectively. This distribution of deletions and duplications can be observed in the cohorts presented by France, China, and Spain, as well as the multi-ethnic cohort. In the current study, deletions comprise 49.0% (386/788) of all mutations, which is significantly lower than the expected 65%. These values correspond to the results of the American study (Flanigan K.M. et al.). As stated previously, this lower rate of deletions in the USA cohort might be explained by a systematic error of patient selection, considering the fact that some patients may have carried minor mutations, which could not be detected for a long time. After this, Sun C. et al. suggested that the differences in DMD/BMD mutation rates could be caused by populational and ethnic diversity. Aside from that, the DMD deletion frequency in the population may vary depending on the geography and race [30]. This was confirmed by multiple researchers from different countries when comparing the mutation spectrum. It is necessary to consider when planning genetic screening, as well as therapeutic procedures, in the exact country [16].

As mentioned above, the location of deletions in the dystrophin gene is not coincidental. It is known that two “hot” regions exist—on the 5′-terminus and in the distal half of the central rod domain, predominantly involving exons 44–53 [14]. We also noted this uneven deletion distribution in the examined cohort. In 62.2% (240/386) of cases, deletions were located in the rod domain, and in 27.2% (105/386) of cases—in the 5′ region (Figure 4). Deletions spanning exons outside of the “hot regions” were encountered quite rarely—in 10.6% (41/386) of all cases. We identified 24 deletions (6.2%) located in exons 21–43 and 17 deletions (4.4%), limited to exons 56–79. Aside from that, one patient had a deletion of the entire gene. It is worth noting that the most common rearrangement in the examined cohort was the deletion of three exons (48–50) detected in 6% of patients (16/386). According to the data presented by other countries, the most common deletion depends on the population, which can be explained by different genetic environments and various exogenic factors [31].

In the examined cohort, duplications comprised 14.5% (114/788) of all mutations. This percentage of patients with duplications differs from the global picture represented by other worldwide cohorts. Unlike the deletions, the majority of duplications were located in the 5′ region—6% (64/114); they were also quite heterogeneous and were identified only once each. The exception was the duplication of exon 2: in the examined cohort, it was detected in 11 patients, which corresponds to the data presented in multiple studies [24,32].

### 3.3. Minor Mutations

Worldwide, minor mutations comprise approximately 18–25% of all mutations [25,33]. The minor mutation analysis was historically a difficult task, mainly because of the enormous size of the gene making it hard to sequence [26]. When comparing the data presented by different countries, we can note that minor mutation rates vary from 18.1% (Spain) to 45.1% (USA) (Table 2). The results obtained in the current study are comparable to the data presented by Flanigan K.M. et al.; the cumulative rate of minor mutations in the Russian Federation is 36.4%, which is significantly higher than the average values. This can be explained by populational and ethnic differences. In the current study, we analyzed the DMD mutation spectrum in patients born between 2018–2021 who applied for molecular genetic diagnostics for the first time (89/788). The group was selected among patients with mutations in the DMD gene (the overall spectrum of mutations is presented earlier in the article). In this group, deletions comprised 50.6% (45/89) of all mutations, duplications—14.6% (13/89), and minor mutations—37.1% (33/89). These results allow us to assume that the mutation spectrum differences are caused by ethnic variability between populations and not by a patient selection bias.

The second most common DMD variant type in Russian patients were nonsense mutations (19.3%), leading to premature stop codon formation and impairing the production of the full-sized protein. The rate of detected nonsense mutations was significantly higher than in worldwide cohorts, in which nonsense mutations cause 10–15% of DMD/BMD cases [34]. Nonsense mutation detection is of utmost importance: since 2020, Ataluren (Translarna) treatment, which leads to restoration of full-sized protein synthesis, is available in the Russian Federation.

Ataluren is an oral medication (derivative of benzoic acid), which allows the translating ribosome to read the information from mRNA containing the premature stop codon and to synthesize full-sized dystrophin [35]. The medication is approved for ambulatory patients over the age of 2 years. Skipping stop codons is the ideal approach to DMD/BMD carrying a nonsense mutation treatment due to the ability to target any region of the DMD gene regardless of the mutation’s location [36]. The intake of the drug leads to a delay in motor function loss, which is vital for the patients, seeing that retaining motor function directly improves the patient’s quality of life [37]. The current study shows that the number of patients eligible for Ataluren treatment is significantly higher than the number of patients eligible for other therapy types (Figure 5).

### 3.4. Pathogenetic Therapy

The DMD mutation spectrum analysis allows us to determine the rates of different variants, which is essential for therapy selection in any particular country. In recent years, many clinical trials aimed at DMD/BMD treatment have been carried out. One of the perspective approaches is therapy using exon skipping. As a result of this treatment, the reading frame is restored due to the exon skipping enabling the production of truncated dystrophin and the replacement of the DMD phenotype with a milder BMD phenotype [38]. This approach uses anti-sense oligonucleotides, which bind to the corresponding DMD pre-mRNA exons before splicing, leading to the skipping of this exon during mRNA processing [39].

As of now, four medications for skipping the exons most commonly undergoing rearrangements are approved for use in the Russian Federation: AMONDYS 45 (Casimersen) for skipping exon 45, EXONDYS 51 (Eteplirsen)—exon 51, and VYONDYS 53 (Golodirsen) and VILTEPSOTM (Viltolarsen) for exon 53.

It is known that worldwide approximately 13% of patients are eligible for therapy based on skipping exon 51 (Eteplirsen), which leads to the reading frame restoration. The use of this medication increases the dystrophin expression and decelerates the disease progression [40]. As of now, this is the largest group of patients eligible for exon skipping therapy [41]. In the examined cohort, they comprise 5.9% of all patients (47/788).

Skipping exon 53 (Golodirsen, Viltolarsen) is applicable for 8–10% of patients worldwide (7.7%, according to Treat NMD) [42]. In the Russian Federation, the proportion of patients eligible for therapy with Viltolarsen and Golodirsen is 4.7% (37/788). Approximately 8% of all DMD patients have mutations suitable for exon 45 skipping [43]; in the examined cohort, these patients comprised 4.3% (34/788) (Figure 5). To conclude, according to the data obtained during the selective DMD screening in the Research Centre for Medical Genetics, the cumulative rate of patients eligible for therapy with skipping the most frequent exons—45, 51, 53—is two times lower (14.9% (118/788)) than the number described in the literature and registered in Treat NMD (28.8%). This can be explained both by the fact that among Russian patients the deletion rate is significantly lower than in previously described cohorts and the specificities of the deletion spectrum in the Russian Federation. The comparison of Russian patients and patients from Treat NMD, which are eligible for therapy based on skipping exon, is presented in Table 3.

According to the data presented in the current study, as of now, the available therapeutic approaches, such as skipping exons most commonly undergoing rearrangements (51, 53, 45) and premature stop codons (Ataluren), are suitable for 34.3% (270/788) of patients. According to the Treat NMD register, the total number of patients eligible for therapy is approximately 39.0%, which is comparable with our results. However, in the Russian cohort the majority of these patients are suitable for Ataluren treatment (19.3%), while in the multi-ethnic cohort they comprise 10.2%.

It is worth noting that all existing medications for DMD/BMD treatment decelerate the disease progression but do not allow the patient to recover the muscular tissue and regain the lost functions; therefore, the therapy effectiveness directly depends not only on the therapy itself, but also from the stage of the pathological process, on which the pathogenetic therapy was initiated [44]. For example, Casimersen treatment (skipping exon 45) is appropriate only for patients, whose ability to move upper limbs independently has not been lost. The published experiment results and patient follow-up showed respiratory function improvement and motor skill loss deceleration [43,45].

Currently, the Astellas Gene Therapies company is carrying out clinical trials aimed at the treatment of DMD patients with exon 2 duplication. It is possible due to the vector delivery of non-coding small nuclear RNA U7 (U7snRNA) modified to bind with acceptor and donor splice sites flanking exon 2 of the DMD gene. U7snRNA prevents the binding of splicing enzymes with pre-mRNA, which excludes the flanked exon from mature mRNA. The testing of this approach on mouse and human cells shows that completely excluding both copies of exon 2 leads to the synthesis of truncated but functionally active dystrophin [46]. Currently, three participants with confirmed duplication of exon 2 (NCT04240314) are undergoing trials [47]. The only downside of this therapy is the inability to skip only one copy of exon 2 because U7snRNA is unable to distinguish identical copies of exon 2, which theoretically would lead to the restoration of the normal transcript. As stated above, the duplication of exon 2 is the most common duplication in multiple studies, including the current study: approximately 1.4% (11/788) in the examined cohort.

## 4. Materials and Methods

### 4.1. Selective DMD/BMD Screening Program in the Russian Federation

Since 1 October 2018, a DMD/BMD selective screening program, aimed at early patient detection, is carried out in the DNA diagnostics laboratory of the Research Centre for Medical Genetics. As of early 2022, patients are being referred from 77 out of 85 regions of the Russian Federation, as well as Kazakhstan, Belarus, Armenia, and Georgia.

The main inclusion criteria are: gender (male), a clinical diagnosis of DMD/BMD, or a significant increase in creatine phosphokinase (CPK) (>2000 U/L). The DMD/BMD clinical diagnosis was verified by clinicians in 184 regional medical institutions throughout Russia.

As of 25 April 2022, 1072 patients from 1025 non-related families underwent all stages of molecular genetic diagnostics based on the selective screening program, and the number of patients continues to increase.

### 4.2. Examination Methods

The biological material was collected in laboratory rooms of medical genetic counseling institutions from various regions of the Russian Federation. Informed consent was obtained from all patients (or their parents/guardians for underage patients). The whole venous blood samples were collected into single-use plastic test tubes containing an anticoagulant (EDTA). DNA was extracted from peripheral blood leukocytes using the Wizard^®^ Genomic DNA Purification Kit (Promega, Madison, WI, USA), according to the manufacturer’s protocol [48].

The first stage of screening was the search for gross deletions and duplications of one or several exons of the DMD gene by means of multiplex ligase-dependent probe amplification (MLPA) using commercially available kits (MLPA SALSA P034 and P035 DMD probemix, MRC-Holland, The Netherlands). The reactions were carried out, according to the manufacturer’s recommendations [49]. The reaction product was detected with fragmentary analysis on ABI Prism 3500 (“Applied Biosystems”). The data were analyzed using the Coffalyser.Net™ software. In case of a single-exon deletion, the results were confirmed with PCR or Sanger sequencing according to the guidelines. At the moment duplications were not confirmed by other methods, according to the guidelines [50]. MLPA has shown the best accuracy for detecting this type of mutation; CGH can skip duplication due to the varying density of probes, real-time PCR often has less sensitivity than MLPA, and genome sequencing to confirm duplication requires excessive resources.

The second stage was a search for minor mutations using a custom NGS panel, consisting of 31 genes. This panel allows to analyze the coding sequences and exon-intronic regions of the DMD gene, as well as the genes associated with genetic forms of various types of limb-girdle muscular dystrophy (CAPN3, EMD, SGCG, SGCA, SGCB, SGCD, TCAP, FKRP, POMT1, POMT2, ANO5, FKTN, ISPD, LMNA, CAV3, DAG1, POPDC3, FHL1, GAA, PLEC, POMGNT1, POMGNT2, GMPPB, HNRNPDL, GNE, FKBP14, DYSF, DNAJB6, BVES, TRIM32) (Attachment 1). The panel sequencing was carried out on a next-generation Ion S5 sequencer (Thermo Fisher Scientific, Waltham, MA, USA).

The detected variants were described, according to the http://varnomen.hgvs.org/recommendations/DNA (accessed on 19 October 2022) (version 20.05) nomenclature [51]. The sequencing data were processed using the ngs-data.ru software [52]. The population frequencies of detected variants were estimated using the cohorts from the “1000 genomes” project and the Genome Aggregation Database (gnomAD v2.1.1). The clinical relevance of the detected variants was evaluated, according to the OMIM database, the HGMD^®^ Professional version 2021.4 pathogenic variant database, the specialized LOVD database (https://databases.lovd.nl/shared/genes/DMD (accessed on 18 September 2022)), and the “Guide to interpretation of human DNA sequence obtained with mass parallel sequencing (MPS)” [21].

In addition, the verification of the detected variant using direct automated Sanger sequencing was carried out for patients with nonsense mutations, as this is a condition for prescribing Ataluren to patients.

## 5. Conclusions

As a result of selective screening, 1137 patients from 1083 families were given the opportunity to undergo free genetic testing. Pathogenic and likely pathogenic variants of the DMD gene were detected in 788 out of 1025 families (76.9%). Causative variants in other genes were detected in 94 non-related patients (9.0%). This high informativity of the selective screening program was made possible due to the molecular diagnostic approach, which includes the detection of gross deletions and duplications of one or several exons of the DMD gene with subsequent minor mutation detection using massive parallel sequencing in patients with a negative MLPA result. Aside from that, the use of a custom NGS panel consisting of 31 genes allows us to identify patients with other muscular dystrophy types, which is vital for their management. The clinical diagnostics of DMD/BMD and other neuromuscular disorders in pediatric and medical genetic counseling centers around various regions of the Russian Federation, contributing to the program’s high informativity.

The examination of the DMD mutation spectrum showed that there are significant differences in frequencies of various mutation types between Russian patients and other cohorts. The deletion rate is slightly lower than the described 65.0%, reaching only 49.0%. The second most common mutation type are point mutations (36.5%); 19.3% are nonsense mutations, which, in fact, makes Ataluren treatment possible for a large number of DMD/BMD patients. Aside from that, therapy based on skipping the exons most commonly affected by rearrangements—51, 53, 45—is available for 14.9% of patients. Cumulatively, the existing therapy types are suitable for 34.3% of patients.

Thus, mutation type identification is of the utmost importance for making a correct diagnosis, prognosis, and individual treatment selection for DMD/BMD patients, as well as for the medical genetic counseling for their families. Considering the fact that DMD/BMD is a rapidly progressing disorder and the muscular tissue destruction is irreversible, it is vital to confirm the molecular genetic diagnosis as early as possible. This would allow us to prescribe effective therapy at an early stage of the disease.

## Figures and Tables

**Figure 1 ijms-23-12710-f001:**
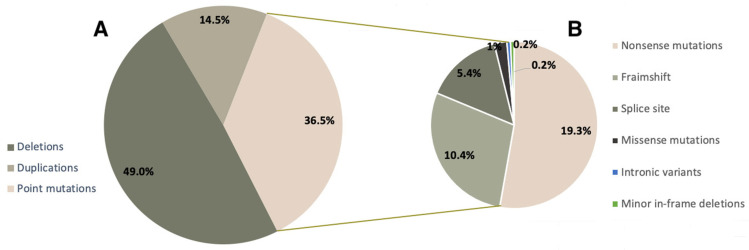
Mutation spectrum of the DMD gene in Russian patients.

**Figure 2 ijms-23-12710-f002:**
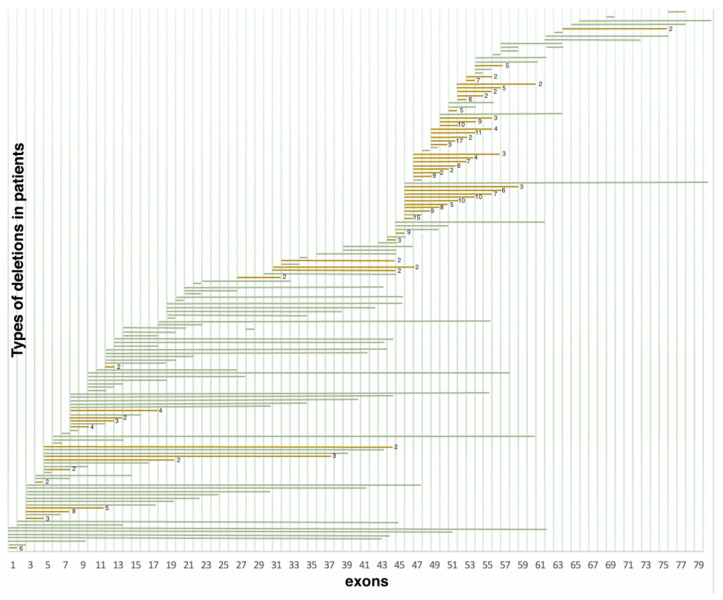
Distribution of deletions in various exons (*n* = 386) (green represents deletions encountered once, yellow represents deletions encountered multiple times).

**Figure 3 ijms-23-12710-f003:**
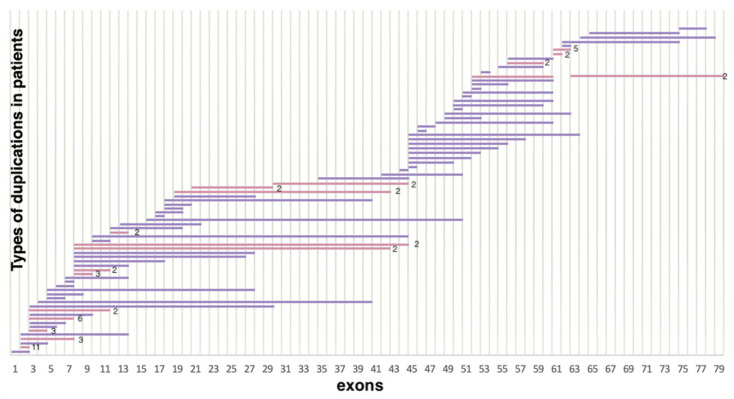
Distribution of duplications in the DMD gene (*n* = 114) (lilac represents duplications encountered only once, pink represents duplications encountered multiple times).

**Figure 4 ijms-23-12710-f004:**
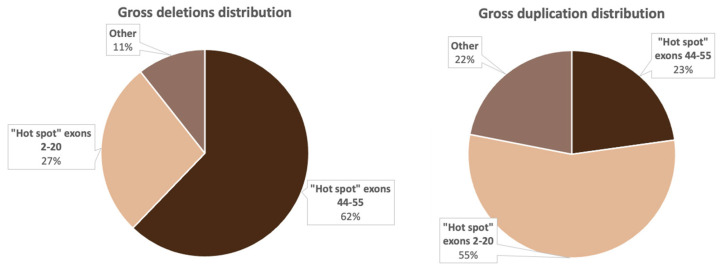
Distribution of the most common deletions and duplications in the DMD gene in Russian patients.

**Figure 5 ijms-23-12710-f005:**
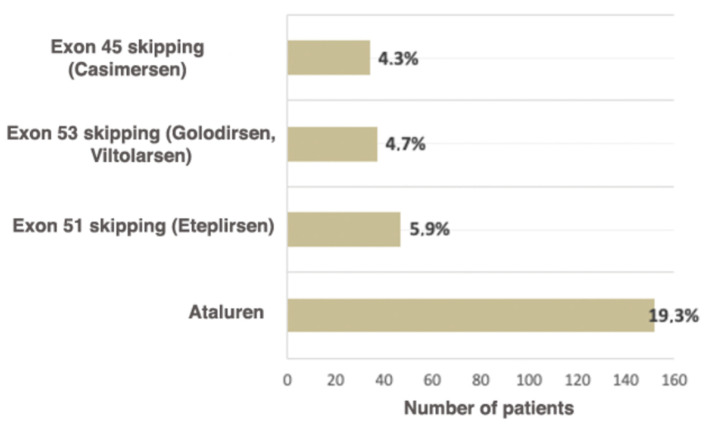
Russian patients eligible for available therapeutic approaches correcting specific mutation types.

**Table 1 ijms-23-12710-t001:** Rates of minor mutations in the DMD gene.

Mutation Type	Number of Patients, *n*	Proportion among Patients with Minor Mutations, %	Proportion among All Patients with Detected *DMD* Variants, %
Nonsense mutations	152	52.7	19.3
Frameshift	81	28.1	10.3
Splice site	43	14.9	5.4
Missense mutations	8	2.7	1.1
Intronic mutations	2	0.8	0.2
Minor in-frame deletions	2	0.8	0.2
TOTAL	288	100	36.5

**Table 2 ijms-23-12710-t002:** Comparison of various DMD mutation rates in DMD/BMD patients from the Russian Federation and other countries.

Mutation Type	TREAT-NMD [1] (*n* = 7145)	Tuffery-Giraud S. et al., France (*n* = 2084) [24]	Kong. X et al., China [25] (*n* = 1051)	Flanigan K.M. et al., USA [18] (*n* = 1111)	Juan-Mateu J. et al., Spain [26] (*n* = 576)	Takeshima Y. et al., Japan [27] (*n* = 442)	Neri M. et al., Italy [16] (*n* = 1902)	Russian Federation (*n* = 788)	*p*-Value *
Deletions (%)	68.5	67.4	70.4	42.9	70.5	61.0	65.0	49.0	**0**
Duplications (%)	11.0	10.3	8.3	11.0	11.1	9.0	9.9	14.5	**0.0024**
Nonsense (%)	10.2	8.9	9.7	26.5	9.4	16.0	10.5	19.3	**0**
Frameshift (%)	6.9	7.1	5.5	11.4	4.5	8.0	7.3	10.3	**0.0004**
Splice site (%)	2.8	6.0	2.8	5.8	3.5	5.0	4.6	5.4	**0**
Missense (%)	0.4	0.3	1.4	1.4	0.7	–	1.9	1.2	**0.0019**
Intronic mutations (%)	0.3	–	–	–	–	1.0	–	0.2	0.76

Note: * obtained as a result of comparison with the TREAT-NMD DMD Global register data. Bold represents statistically significant differences (*p* < 0.05).

**Table 3 ijms-23-12710-t003:** Comparison of the numbers of patients eligible for exon skipping therapy in the Russian Federation and in the multi-ethnic cohort.

Exon for Skipping	Number of Patients in the Russian Federation, *n*	Rate among All Patients with *DMD* Mutations (*n* = 788), %	Rate among All Patients with Deletions (*n* = 386), %	Rate among All Patients with *DMD* Mutations *, % (Data from Treat NMD)	Rate among All Patients with Deletions *, % (Data from Treat NMD)
51	47	5.9	12.2	13.0	19.1
53	37	4.7	9.6	7.7	11.4
45	34	4.3	8.8	8.1	11.8
TOTAL	118	14.9	30.6	28.8	42.3

Note: * data from the TREAT-NMD DMD Global register.

## Data Availability

Not applicable.

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
