# Peer review of "Specificities of the DMD Gene Mutation Spectrum in Russian Patients"

_ijms, 2022, doi:10.3390/ijms232112710_

Round 1
Reviewer 1 Report
The authors, with this work, aimed to report the DMD mutation spectrum in Russia and to increment the diagnosis among Russian children. Both of these aims are important in child treatments and in the field knowledge increment.
The work is well presented and the figures are informative, though the pictures themselves have poor quality.
My principal and bigger concern for this paper is the literature review. I found it very approximative because of the big mistake in the paper/country association (see the pdf file) that is perpetuated all over the discussion and results and leads the authors to some, of course wrong, assumptions. Moreover, I think that the authors' aim and the manuscript design definitely request a larger and in-deep literature review. My suggestion is to reshape the discussion substantially, including more papers to use a comparison, especially the ones that consider populations genetically nearest to the Russian.
Considering the vastity of Russian territory and the different ethnicity that populate the territory, I think it could be interesting to divide the mutations found among the ethnic subgroups.
I have also some concerns about the validation of mutations because, if I understood correctly, only the single exon deletions and nonsense mutations have been validated.

Author Response
We are very grateful to you for the attentive review of our work. Your questions prompted us to search for additional information. Unfortunately, we could not fulfill all your wishes, but we tried our best.
Point 1: My principal and bigger concern for this paper is the literature review. I found it very approximative because of the big mistake in the paper/country association (see the pdf file) that is perpetuated all over the discussion and results and leads the authors to some, of course wrong, assumptions. Moreover, I think that the authors' aim and the manuscript design definitely request a larger and in-deep literature review. My suggestion is to reshape the discussion substantially, including more papers to use a comparison, especially the ones that consider populations genetically nearest to the Russian.
Response 1: We have corrected the mistake in the paper/country associations. Also we have added some papers to use a comparison. Thank you for the article suggestions, it was very helpful for discussion in the article. Some data has been entered into the table 2. All our changes are highlighted in green in the text.
Point 2: Considering the vastity of Russian territory and the different ethnicity that populate the territory, I think it could be interesting to divide the mutations found among the ethnic subgroups.
Response 2: Thank you for your comment. At the moment, the ethnicity of patients in the study are predominantly Russians. But it would be interesting to see the spectrum of mutations by regions. We will try to divide the mutations found among the ethnic subgroups and regions in a future article.
Point 3: I have also some concerns about the validation of mutations because, if I understood correctly, only the single exon deletions and nonsense mutations have been validated.
Response 3: Thank you for the comment. First of all, we checked a single-exon deletions with PCR or Sanger sequencing, because they often mask minor mutations. Validation of nonsense mutations is carried out in our center, as this is a condition for prescribing Ataluren to patients. At the moment duplications weren’t confirmed by other methods. MLPA has shown the best accuracy for detecting this type of mutation - CGH can skip duplication due to the varying density of probes, real-time PCR often has less sensitivity than MLPA and genome sequencing to confirm duplication requires excessive resources.
Reviewer 2 Report
In the article “Specificities of the DMD gene mutation spectrum in Russian patients” Zinina and coworkers studied the distribution of the DMD gene mutation in 788 Russian families, which is very important for the DMD/BMD patients in order to identify the right therapy available for its mutation. The article is well written and presented. However, before the publication in International Journal of Molecular Sciences, the authors should address some modifications.
· The phrase in the introduction “The most common mutations in case of DMD/BMD are gross deletions and duplications of one or several exons” is repeated twice in 10 lines, in my opinion should be left just the second.
· In figure 1 specify A for the first pie chart and B for second one, to explain better in the text to which the authors refer to.
· Please write extensively MLPA in the first line of the Results Section 2.1 Deletions, since it is explained in the section Methods which is after.
Suggestions:
· I would like to suggest to insert in the section Introduction a small description of the role of the protein dystrophin in the DGC (Dystrophin-glycoprotein complex) to understand better the role of this protein and why its absence translates into this two pathologies.
Author Response
Point 1: The phrase in the introduction “The most common mutations in case of DMD/BMD are gross deletions and duplications of one or several exons” is repeated twice in 10 lines, in my opinion should be left just the second.
Response 3: Thank you for a note! We corrected that.
Point 2: In figure 1 specify A for the first pie chart and B for second one, to explain better in the text to which the authors refer to.
Response 2: Thank you for the suggestion. We added it.
Point 3: Please write extensively MLPA in the first line of the Results Section 2.1 Deletions, since it is explained in the section Methods which is after.
Response 3: This phrase has been corrected according to your notes, thank you.
Point 4: I would like to suggest to insert in the section Introduction a small description of the role of the protein dystrophin in the DGC (Dystrophin-glycoprotein complex) to understand better the role of this protein and why its absence translates into this two pathologies.
Response 4: We agree with you, thank you for comment. All the necessary information has been added to the article.
Round 2
Reviewer 1 Report
Thanks to the authors for having amended the paper following my advice. For future work, I would like to suggest that European guidelines for DMD diagnosis highly recommend validating the MLPA results also in duplications. For reference please consider page 1143 of EMQN best practice guidelines for genetic testing in dystrophinopathies. Fratter C, Dalgleish R, Allen SK, et al. 2020, Eur J Hum Genet, Vol. 28, p. 1141–1159.
Author Response
We thank the Reviewer for thoroung analysis of the manuscript and useful suggestions which hopefully allowed us to improve the manuscript. We will consider your suggestion for our future work.